# Increased Glomerular Filtration Rate in Early Stage of Balkan Endemic Nephropathy

**DOI:** 10.3390/medicina55050155

**Published:** 2019-05-17

**Authors:** Ljubica Djukanović, Višnja Ležaić, Danica Bukvić, Dušan Mirković, Ivko Marić

**Affiliations:** 1Faculty of Medicine, University of Belgrade, 11000 Belgrade, Serbia; visnjalezaic@gmail.com; 2Department of Nephrology, Clinical Centre of Serbia, 11000 Belgrade, Serbia; 3Special Hospital for Endemic Nephropathy, 11550 Lazarevac, Serbia; danicabukvic@gmail.com (D.B.); nefropatija@eunet.rs (I.M.); 4Faculty of Pharmacy, University of Belgrade, 11000 Belgrade, Serbia; mirkovicd@mts.rs; 5Centre for Medical Biochemistry, Clinical Centre of Serbia, 11000 Belgrade, Serbia

**Keywords:** Balkan endemic nephropathy, creatinine clearance, cimetidine, iohexol clearance

## Abstract

*Background*: A previous study indicated that Balkan endemic nephropathy (BEN) patients in the early stage of the disease had significantly higher creatinine clearance (Ccr) than healthy persons. The aim of the study was to assess whether tubular creatinine secretion affects Ccr in early stages of BEN and to check the applicability of serum creatinine-based glomerular filtration rate (GFR) equations in these patients. *Methods*: The study involved 21 BEN patients with estimated GFR (eGFR) above 60 mL/min/1.73 m^2^, excluding any conditions that could affect GFR or tubular creatinine secretion, and 15 healthy controls. In all participants Ccr with and without cimetidine and iohexol clearance (mGFR) were measured and eGFR calculated using Chronic Kidney Disease Epidemiology Collaboration (CKD-EPI) and Modification of Diet in Renal Disease Study (MDRD) equations. Glomerular hyperfiltration cutoff (GFR-HF) was calculated. *Results*: There was no significant difference between the groups in Ccr before and after cimetidine or for eGFR, but mGFR was significantly higher in BEN patients than in controls (122.02 ± 28.03 mL/min/1.73 m^2^ vs. 101.15 ± 27.32 mL/min/1.73 m^2^; *p* = 0.032). Cimetidine administration reduced Ccr by 10% in both groups. The ratio of Ccr to mGFR was significantly above one in seven BEN patients and five controls and their mGFR values were similar. Seven other patients and eight controls had this ratio equal to one, while values below one were recorded for seven more patients and two controls. mGFR of all these 14 patients was significantly higher than that of healthy controls (129.88 ± 27.52 mL/min/1.73 m^2^ vs. 107.43 ± 19.51 mL/min/1.73 m^2^; *p* = 0.009). Mean GFR-HF was significantly higher than mGFR in controls, but these two values were similar in BEN patients. eGFR underestimated mGFR in both BEN patients and controls. *Conclusion*: The ratio of Ccr to mGFR and mGFR to GFR-HF indicated that elevated mGFR in early stages of BEN could be explained by increased glomerular filtration, but tubular creatinine secretion augmented Ccr in a smaller proportion of patients, who did not differ from healthy subjects.

## 1. Introduction

Balkan endemic nephropathy (BEN) is a familial, chronic tubulointerstitial kidney disease with insidious onset and a slow progressive asymptomatic course. Tubular disorders, particularly tubular proteinuria is a kidney function disorder that occurs at the very beginning of BEN [1,2] and is recommended as the main diagnostic criterion of BEN [3]. However, we previously noticed that in the early stage of the disease BEN patients had significantly higher creatinine clearance (Ccr) than individuals with other kidney diseases and healthy control subjects [4]. This finding was not checked later. While Ccr is a measure of glomerular function, it cannot be overlooked that creatinine is excreted not only by glomerular filtration but also by proximal tubular secretion [5]. The question arises as to whether this previously noticed increased Ccr in the early stage of BEN truly results from higher glomerular filtration or from increased tubular secretion of creatinine. Searching for an answer to this question we examined the effect of cimetidine on 24-h Ccr in BEN patients in the early stage of the disease, as cimetidine is known to block secretion of creatinine in proximal tubules [6,7,8]. In addition, clearance of iohexol was measured, which enabled reliable determination of glomerular filtration rate (GFR) [9]. The aim of the study was to assess the contribution of tubular creatinine secretion on Ccr in early stage BEN and to check the applicability of serum creatinine-based GFR equations in these patients.

## 2. Materials and Methods

### 2.1. Study Groups

The study involved 21 patients with BEN and 15 healthy controls. BEN participants were selected from the population of patients detected during screening studies in the Kolubara region and BEN was diagnosed using previously defined criteria [3,10]. Forty individuals with BEN were invited for examination in the Special Hospital for Endemic Nephropathy in Lazarevac, 32 of them responded and 27 persons agreed to participate in the study. Exclusion criteria were: GFR estimated by MDRD equation below 60 mL/min/1.73 m^2^, body mass index (BMI) above 30 kg/m^2^, pregnancy, diabetes, a history of allergy to iodine and cimetidine, use of H2-blockers, antiacids or drugs that are known to interfere with creatinine secretion. Finally, 21 patients with BEN met the criteria and were included in the study. The control group consisted of fifteen healthy persons among whom ten were potential kidney donors examined for suitability while the remaining five were persons sent to the hospital for evaluation of kidney function in whom kidney disease was excluded. The Ethics Committee of the Clinical Centre of Serbia approved the study (No: E-KCS/KS 025/2014; 2/12/2014), and both patients and healthy controls gave their informed consent.

### 2.2. Study Protocol

All persons who accepted to participate in the study were given recommendation to collect 24-h urine and bring it to the first examination after a 12-h fast (Scheme 1). On that occasion, an anamnesis was taken and the objective examination included measurement of blood pressure, body weight and height. BMI was calculated according to the formula: weight (kg)/height^2^ (m^2^) and body surface area (BSA) from the equation of DuBois and DuBois [11]. Volume of 24-h urine was measured and an aliquot taken for creatinine measurement. Also, a blood sample was taken for determination of creatinine as well as some fresh urine for measurement of creatinine, protein, albumin and alpha-1-microglobulin. All participants were given instructions about nutrition (abstain from high protein foods), fluid intake (much fluid with no caffeine) and avoidance of medication (nonsteroidal anti-inflammatory drugs, diuretics) that should be followed the day before and during the cimetidine and iohexol clearance tests. The participants received instructions for the two-day cimetidine test which they performed at home. The first day they took three doses of cimetidine orally (800 mg every 8 h beginning at 7 a.m.). During the second day they collected the 24-h urine which they brought the following day to the hospital. The volume of the collected urine was measured, and an aliquot taken for determination of creatinine concentration. Also, a venous blood sample was taken for creatinine measurement as well as a basal blood sample for iohexol clearance. A bolus of 5 mL of 300 mg/mL iohexol (Omnipaque 300, GE Healthcare, Marlborough, MA, USA) was administered into an arm vein and flushed in with 10 mL of normal saline. Venous blood was taken from the contralateral arm after 120, 180, 240, and 360 min. The blood samples were centrifuged, the serum separated and stored at −20 °C until assayed.

### 2.3. Laboratory Analyses

Serum and urine concentrations of creatinine were determined enzymatically using the creatinine PAP (CREA) enzymatic colorimetric test kit (Boehringer Mannheim GmbH, Mannheim, Germany). The characteristic reaction sequence were achieved with creatininase, creatinase, sarcosine oxidaze and peroxidaze.

Serum iohexol concentration was measured by reverse-phase high-performance liquid chromatography (C18HPLC column). 100 µL of serum was mixed with 100 µL 5% HClO_4_, and centrifugated at 10,000× *g*. 20 µL of the supernatant was injected in the HPLC system which consisted of autosampler “AS-100 Bio-Rad” (Bio-Rad Laboratories, Hercules, CA, USA), spectrophotometric, UV/VIS detector “Lambda-Max 481” (LabMakelaar Benelux BV, Zevenhuizen, The Netherlands), and isocratic HPLC pump, “Bio-Rad 1350 T” (Bio-Rad Laboratories). All data were processed utilizing chromatographic program “Chrom-Line 4.20, Bio-Rad”. Chromatographic separation of the two enantiomeric iohexol forms was carried out isocraticaly, using a mobile phase of acetonitrile-water (60:40 v/v), on HPLC column “LiChroshare 60” (1504.6 mm, 5µm) at a flow rate 1 mL/min. Detection has been done at 254 nm. The intra- and interassay coefficients of variation for the HPLC assay were 2.7 and 6% respectively.

Urine protein was determined colorimetrically with pyrogallol red (normal value < 20 mg/mmol creatinine), while an immunonephelometric assay (BN II nephelometer, Dade Behring Marburg, Germany) was used for albumin (normal value < 3.4 mg/mmol creatinine) and alpha-1-microglobulin (normal value < 1.5 mg/mmol creatinine).

GFR was determined as follows:
by calculation of the 24-h creatinine clearance before (Ccr) and after administration of cimetidne (CcrC) using the standard formula and recalculated on BSA;by calculation of iohexol clearance (measured GFR-mGFR) as described by Schwartz et al. [12];by estimation with the MDRD (eGFR-MDRD) and CKD-EPI (eGFR-CKD-EPI) [13] equations using serum creatinine levels.


A ratio of Ccr to mGFR equal to one indicates creatinine glomerular filtration. Mean Ccr/mGFR for the BEN group was 1.07 (95% confidence interval: 0.96–1.17) and we considered each value in the range from 0.96 to 1.17 as equal to one.

In addition, the glomerular hyperfiltration cutoff (GFR-HF) was calculated using the following definition of hyperfiltration: mGFR > 144 − (age – 40), i.e., the cut-off takes into account progressive reduction of GFR by 1 mL/min/1.73 m^2^ for every year after the age of 40 years [14].

### 2.4. Statistical Analysis

All analyses were performed using the SPSS software package (Version 10; SPSS. IBM Corp. Released 2012. IBM SPSS Statistics for Windows, Version 21.0. Armonk, NY: IBM Corp). Data are expressed as frequency, mean values and standard deviations or as medians and interquartile ranges depending on the characteristics of the variable. Comparison of the variables between groups was made using the Student-t test, Kruskal–Wallis test, chi-square test or Fisher’s test, as appropriate. Correlation was analyzed through the Pearson correlation coefficient.

The performance of GFR estimated with the CKD-EPI and MDRD equations was also assessed. Bias was defined as the difference between eGFR and mGFR and the median and 95% confidence interval (CI) for the mean are presented. The interquartile range of the difference, obtained as the difference between 75th and 25th percentiles, was used as a measure of precision, while accuracy was expressed as P30 defined as the percentage of eGFR differing by not more than 30% from mGFR.

## 3. Results

The main characteristics of the two groups examined are presented in Table 1. Patients with BEN had higher mean blood pressure which was within the normal range in the control group. Also, urinary excretion of the three examined proteins was significantly higher in BEN patients than in healthy controls.

The values for GFR determined by different methods are presented in Table 2. Although Ccr before and after cimetidine use was higher in BEN patients than in controls, the differences were not significant. The difference in mGFR, which was higher in BEN patients than in healthy controls, was statistically significant. The administration of cimetidine reduced Ccr by 10% on average both in BEN patients and healthy controls but the changes were not statistically significant (Table 2). Accordingly, the mean ratio between Ccr without cimetidine and mGFR was about 1.1, but the ratio between Ccr with cimetidine and mGFR was 1.01 in both groups examined.

Mean GFR-HF, calculated as the cutoff value of hyperfiltration with BSA normalization and age adjustment, was similar to mGFR in BEN patients, but significantly higher in the healthy controls (*p* = 0.0005).

A ratio of Ccr to mGFR equal to one indicates glomerular filtration of creatinine, but a ratio higher than one points to net renal creatinine secretion. As mean Ccr/mGFR for the BEN group was 1.07 (95% confidence interval:0.96–1.17), we considered each value from 0.96 to 1.17 as a Ccr/mGFR ratio equal to one. In order to point out the association between tubular transport of creatinine and mGFR, we presented the relationship between the ratio of Ccr to mGFR and mGFR in both BEN patients and healthy controls (Figure 1). It can be seen that a similar proportion of BEN patients (7/21) and healthy persons (5/15) had Ccr/mGFR above 1.17 and their mGFR values were similar (108.49 ± 20.25 mL/min/1.73 m^2^ vs. 99.18 ± 23.92 mL/min/1.73 m^2^; *p* = 0.221). In 7/21 BEN patients and 8/15 controls Ccr/mGFR was equal to one, but in 7/21 BEN patients and 2/15 controls the ratio was below one. mGFR values of all these 14 BEN patients were significantly higher than in 10 healthy persons (129.88 ± 27.52 mL/min/1.73 m^2^ vs. 107.43 ± 19.51 mL/min/1.73 m^2^; *p* = 0.009).

The mean value of eGFR-CKD-EPI was significantly lower in the BEN patient group (*p* = 0.0001) but insignificantly in the control group (*p* = 0.124) in comparison with mGFR of the respective group. The same was found for eGFR-MDRD in patients (*p* = 0.0001) and controls (0.071). Bias (median difference eGFR-mGFR; 95%CI), precision (IQR of the difference) and accuracy (P30) of eGFR compared to mGFR were calculated for both groups. Bias (−44.3; 54.9–−31.9 mL/min/1.73 m^2^) and precision (−65.8–−23.2 mL/min/1.73) of the CKD-EPI equation in BEN patients differed significantly from bias (−8.5; 42.5–2.8 mL/min/1.73 m^2^; *p* = 0.024) and precision (−52–−8.5) of healthy persons but not accuracy (42.9% vs. 46.7%). Similar results were obtained for performance of the MDRD equation: BEN patients–bias (47.5; −58.9–−37.1 mL/min/1.73 m^2^), precision (−69.4–−29.1 mL/min/1.73 m^2^), accuracy (38.1%); healthy controls–bias (17.6; 45.9–0.7; *p* = 0.047), precision (−51.8–−2.4), accuracy (39.7%). On the other hand, agreement between Ccr and mGFR was much closer: BEN–bias (8.6; −6.6–19.1 mL/min/1.73 m^2^), precision (−12–26.9 mL/min/1.73 m^2^), accuracy (78.2%); healthy controls–bias (4; −0.4–24.5), precision (−1.8–25.9), accuracy (78.6%).

Analyzing correlations between all examined variables we found a statistically significant positive correlation between GFR values determined by the different methods both for the BEN group (p: 0.008–0.014) and the control group (p: 0.0001–0.047). There was also a significant negative correlation between GFR values determined by all methods used and patient age (p: 0.0001–0.022), but not for the healthy controls. Figure 2 presents the correlation between mGFR and age for both examined groups.

## 4. Discussion

In order to check whether tubular secretion of creatinine affects Ccr in early stage of BEN, 24h-Ccr was measured before and after cimetidine administration and compared with mGFR in BEN patients and healthy controls. mGFR in BEN patients was significantly higher than for healthy controls, while Ccr values before and after cimetidine differed insignificantly. Moreover, administration of cimetidine reduced Ccr by 10% on average in both groups and the mean ratio of Ccr without cimetidine to mGFR was about 1.1 in both cases. About a third of the participants from both groups had a Ccr to mGFR ratio of above one, and their mGFR were similar. In BEN patients with a Ccr to mGFR ratio equal to and below one, mGFR values were markedly higher than in healthy persons, and similar or even higher than GFR-HF values indicating that increased Ccr in these BEN patients was the result of increased glomerular filtration but not due to greater creatinine secretion.

In a previous study that included 84 BEN patients with normal kidney function, we found 37 (44%) patients with 24-h Ccr (not normalized to BSA) above 150 mL/min and this percentage was significantly higher than in patients with other kidney diseases and healthy controls [4]. In the present study 24-h Ccr with BSA normalization was calculated and the mean value was somewhat higher in BEN patients than in healthy controls but not significantly. However, mGFR was significantly higher in BEN patients than in the controls, although BEN patients were older. The age-related decline of GFR is well documented and the 95th percentile of GFR for persons about 60 years of age is around 90 mL/min/1.73 m^2^ [15,16]. The mean age of our BEN patients was 60.5 years, but 12/21 were older than 60 years and their mGFR was 122.02 ± 28.03 mL/min/1.73 m^2^. As expected, a significant negative correlation was found between age and mGFR in BEN patients. When the correlation between mGFR and age was compared between the groups, it was found that many mGFR values of BEN patients were higher than those for healthy individuals of the same age. The regression line obtained with the mGFR values of BEN patients predicted mGFR of 124 mL/min/1.73 m^2^ for patients aged 60 years, which is noticeably higher than the average value found for healthy persons of the same age [15,16].

There are few reports presenting GFR in BEN patients with normal kidney function measured by reliable and precise methods [17,18,19] and by 24-h Ccr [20]. It is difficult to compare the data because the criteria for classifying patients into groups are different, but nevertheless those studies that involved a larger number of BEN patients with normal kidney function reported Ccr values similar to those found here [18,20]. The question was raised whether the higher values of Ccr in BEN patients with normal kidney function were due to increased tubular secretion of creatinine or to enhanced glomerular filtration. A ratio of Ccr to mGFR equal to one indicates creatinine glomerular filtration, but ratio higher than one point to net renal creatinine secretion [8]. In the present study Ccr/mGFR above one was found in 7/21 BEN patients and 5/15 healthy persons. No significant difference was found in mean mGFR between the subgroups and the Ccr/mGFR ratio was about 1.3 for both. This suggested that tubular secretion of creatinine, which increased Ccr by about 30%, existed in the same proportion of BEN and control participants.

Meanwhile, 7/21 BEN patients and 8/15 healthy controls had ratio of Ccr to mGFR equal to one. Mean mGFR of these BEN patients (120.92 ± 28.37 mL/min/1.73 m^2^) was significantly higher than that of the controls (103.76 ± 19.05 mL/min/1.73 m^2^; *p* = 0.0496), even though the BEN patients were older (59.43 ± 8.67 years vs. 51.86 ± 6.03 years; *p* = 0.007). This indicates that the higher mGFR in this subgroup of patients was due to increased glomerular filtration. Therefore, we calculated the glomerular hyperfiltration cutoff (GFR-HF) using the previously proposed definition of hyperfiltration that takes into account age and BSA [14]. BEN patients with a Ccr to mGFR ratio equal to one had mGFR similar to GFR-HF (124.6 ± 9.3 mL/min/1.73 m^2^) giving an mGFR to GFR-HF ratio of 0.96, while this ratio for the controls was 0.79. This supports our above-stated view that the higher mGFR values in the subgroup of BEN patients with Ccr/mGFR equal to one were due to glomerular hyperfiltration.

In 7/21 BEN patients the Ccr to mGFR ratio was below one (0.65–0.95) indicating tubular reabsorption, but only two healthy persons had such ratios (0.88 and 0.92). Although creatinine is mainly excreted by glomerular filtration and tubular secretion, tubular reabsorption has been found in healthy newborns and premature babies, in dehydrated adults and in healthy old patients [21,22]. In our study all persons were well hydrated during the cimetidine and iohexol tests, and only two BEN patients with Ccr/mGFR below one were more than 70 years old. Although the mean age of the seven patients with Ccr/mGFR below one was 61.1 (range: 40–75), contrary to our expectation, mean mGFR was 138.8 ± 23.4 mL/min/1.73 m^2^, which was higher than their GFR-HF (122.9 ± 11.2 mL/min/1.73 m^2^). This could indicate that increased glomerular filtration existed in all BEN patients with Ccr/mGFR ratio equal to or below one.

Glomerular hyperfiltration occurs in some physiological states (pregnancy, high protein load), in various diseases (diabetes, sickle cell disease, autosomal dominant polycystic kidney disease, secondary focal segmental glomerulosclerosis, chronic kidney disease) and also in persons with obesity or metabolic syndrome, and amongst Indigenous Australians [14,23,24]. All these conditions were avoided by applying our exclusion criteria and preparing patients properly for the GFR measurements and therefore the high mGFR in our BEN patients cannot be explained by the existence of any of the above states. Glomerular hyperfiltration may be due to hemodynamic or tubular changes. The former are caused by changes in afferent-efferent arteriole tonus which is regulated by different vasoactive substances and the latter by increased sodium reabsorption in the proximal tubule which leads to afferent arteriolar vasodilatation and GFR rise [23,25,26]. The pathophysiology of BEN is not sufficiently well investigated but it is considered to be a tubulointerstitial disease whose main cause is intoxication with aristolochic acid [27]. Although the epithelial cells of proximal tubules were considered to be the primary site of aristolochic acid toxicity, recent studies have indicated that changes in endothelial cells of the interlobular and afferent arterioles precede the tubular changes [28,29,30]. It was postulated that the microvascular injury and imbalances in endothelial vasoactive mediators contribute to development of tubular necrosis, apoptosis and fibrosis with consequent renal dysfunction in chronic aristolochic acid nephropathy. Changes in pre- and postglomerular blood vessels have already been described in BEN [31,32]. Thus, Sindjić [32] pointed out that microvascular changes played an important if not the most important role in the pathogenesis of BEN. Such microvascular changes, most likely associated with imbalances in vasoactive substances, might explain the high mGFR found here in almost two-thirds of examined BEN patients. Nine of our BEN patients used ACEI for hypertension treatment. Although their mean mGFR was somewhat lower than for those not taking ACEI (111.72 ± 21.45 mmHg vs. 127.22 ± 29.48 mmHg), the difference was not statistically significant. Moreover, the patients using ACEI were older than the others (65.44 ± 5.66 mmHg vs. 56.75 ± 13.76 mmHg). Nevertheless, this suggested that the role of angiotensin in the occurrence of high GFR in early stages of BEN should be examined, not only to clarify the pathogenesis of this disorder, but also because such treatment might slow down progression of the disease.

On the other hand, the cause of creatinine tubular reabsorption indicated by our results can only be speculated on. As tubular reabsorption of creatinine was taken to be a consequence of immature tubular structure in newborns or tubular changes in the aging kidney, it could be hypothesized that the tubular changes characteristic for BEN lead to creatinine reabsorption at levels corresponding to that described in the elderly [22].

It is well known that GFR estimated by equations using serum creatinine level has insufficient accuracy for subjects with normal kidney function [33,34]. Our findings for Ccr and its ratio to mGFR indicate that this inaccuracy is even more pronounced in BEN patients. Comparison of the performance of eGFR calculated by the CKD-EPI and MDRD equations showed that both of them underestimated true GFR. Their bias was similar and significantly higher than that for healthy controls. Therefore, GFR estimated by these equations cannot be used for evaluation of GFR in BEN patients with normal kidney function but the accuracy of 24-h Ccr is acceptable for routine work.

The main shortcomingof our study is the small number of BEN patients, which is partially due to the small number of patients in the initial stage of the disease now living in endemic villages. Namely, in these villages the elderly have mostly remained, while the young have moved to cities and due to the asymptomatic course of BEN do not consult doctors in the early stage of the disease. Regardless of the small number of patients examined here, for the first time we have compared mGFR with Ccr before and after cimetidine administration and have indicated how much tubular processes and how much glomerular hyperfiltration affect creatinine clearance in the early stages of BEN. The results presented indicated glomerular hyperfiltration in two thirds of our patients and these data are worth checking in a study involving a larger number of subjects from different BEN foci.

## 5. Conclusions

The BEN patients in the early stage of the disease examined here had higher mGFR than healthy persons of similar age and higher values for mGFR than would be expected at that age. Comparison mGFR with Ccr without and with cimetidine administration indicated that tubular secretion of creatinine increased Ccr by about 30%, in one third of the examined BEN patients and healthy controls. The elevated mGFR found in most of the remaining BEN patients could be explained by increase glomerular filtration based on the ratio of Ccr to mGFR. As these results were obtained by examining a small number of patients, it would be worthwhile to check the data on glomerular hyperfiltration in the early stage of BEN in a multicenter study involving a larger group of patients. This is desirable not only to clarify the pathogenesis of BEN, but also to be a guide for the choice of therapy in the early stage of the disease.

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
