# Peer review of "Increased Glomerular Filtration Rate in Early Stage of Balkan Endemic Nephropathy"

_medicina, 2019, doi:10.3390/medicina55050155_

Round 1
Reviewer 1 Report
The authors assess whether tubular creatinine secretion affects Ccr in early stages of BEN and to check the applicability of serum creatinine-based glomerular filtration rate (GFR) equations in BEN patients.The study is straight-forward but a small cohort size is the major limitation of the study. The authors should consider following points to improve the manuscript:
The manuscript title needs to be refined and clear for the readers.
The introduction needs to be improved with appropriate choice of words. The manuscript lacks coherence and adequate background literature.
A graphical design of the undertaken study would improve clarity.
Describe all the methods (Laboratory analysis) in details.
Cohort size is small to make conclusive findings.
The manuscript should include most recent references with respect to the study.
Author Response
We really appreciate valuable comments of reviewer that help us to improve our manuscript. We tried to revise our paper according to reviewer’s comments:
1. The title of the manuscript is changed.
2. The introduction has been rewritten and it seems to me that it is now clearer.
3. In accordance to suggestion of reviewer we plotted Scheme 1 in which the course of participant examination is shown.
4. In order to meet the comment of reviewer, the methods of laboratory analyses are described in details.
5. The reviewer noted that because of a small group of patients it is not possible to make conclusive findings. Therefore, we mitigated the conclusion and emphasized the need for further research (Conclusion at the end of the manuscript).
6. Unfortunately, we could include only a few more recent references with respect to the study, because majority of published papers are devoted to etiology of Balkan nephropathy but not to its pathogenesis.
Reviewer 2 Report
The authors of the study attempted to investigate whether tubular secretion of creatinine serves as a driving force in the hyperfiltration observed in Balkan endemic nephropathy (BEN) patients. Although the study was conducted in a small number of patients, findings from this study indicate a clear dichotomy that exists among the BEN population. For example, seven BEN patients and 1 healthy individual had a ratio of Ccr to mGFR less than one. Nevertheless, there is also a consistent increase in the mGFR in early stages of BEN, which hints that elevated glomerular filtration instead of tubular creatinine secretion, plays a major role. Expanding this study with more number of patients would greatly increase the fidelity of the findings from this study.
Author Response
We thank the reviewer for his opinion. As a lack of the study we identified a small number of patients examined. In the revised manuscript we added that it would be worthwhile to continue the study involving a larger number of subjects from different BEN foci (last paragraph of Discussion and the end of Conclusion).
Round 2
Reviewer 1 Report
The authors have addressed all the concerns to improvise the manuscript.